# The Role of IL-18 in P2RX7-Mediated Antitumor Immunity

**DOI:** 10.3390/ijms24119235

**Published:** 2023-05-25

**Authors:** Serena Janho dit Hreich, Paul Hofman, Valérie Vouret-Craviari

**Affiliations:** 1Faculty of Medicine, Université Côte d’Azur, CNRS, INSERM, IRCAN, 06108 Nice, France; serenahreich@gmail.com; 2IHU RespirEREA, Université Côte d′Azur, 06108 Nice, France; hofman.p@chu-nice.fr; 3FHU OncoAge, 06108 Nice, France; 4Laboratory of Clinical and Experimental Pathology and Biobank, Pasteur Hospital, 06108 Nice, France; 5Hospital-Related Biobank, Pasteur Hospital, 06108 Nice, France

**Keywords:** P2RX7, IL-18, cancer

## Abstract

Cancer is the leading cause of death worldwide despite the variety of treatments that are currently used. This is due to an innate or acquired resistance to therapy that encourages the discovery of novel therapeutic strategies to overcome the resistance. This review will focus on the role of the purinergic receptor P2RX7 in the control of tumor growth, through its ability to modulate antitumor immunity by releasing IL-18. In particular, we describe how the ATP-induced receptor activities (cationic exchange, large pore opening and NLRP3 inflammasome activation) modulate immune cell functions. Furthermore, we recapitulate our current knowledge of the production of IL-18 downstream of P2RX7 activation and how IL-18 controls the fate of tumor growth. Finally, the potential of targeting the P2RX7/IL-18 pathway in combination with classical immunotherapies to fight cancer is discussed.

## 1. Introduction

Current treatments in cancer not only target the cancerous cells, but also take advantage of the composition of the tumor microenvironment (TME) that include immune and stromal cells that have been shown to impact tumor growth [1,2]. Indeed, boosting the antitumor activity of the immune system using blocking antibodies targeting immune checkpoints, such as PD-L1 and CTLA-4, has markedly increased patients’ survival [3,4,5]. Despite the efficacy of immunotherapies and the vast diversity of therapies in cancer, the majority of cancer patients remain resistant to treatments due to innate or acquired resistance mechanisms. Therefore, new strategies to fight this disease are still needed.

The TME is characterized by high extracellular ATP (eATP) that are released from dying cancer cells or immune cells, as reviewed in [6,7]. Such levels of eATP activate the purinergic P2RX7 receptor that is expressed on all cells in the TME [6]. Since ATP is normally present inside the cell, eATP constitutes a damage-associated molecular pattern (DAMP), enabling the recruitment and activation of immune cells into the TME, but can also be recognized by P2RX7-expressing non-immune cells, such as tumor cells. Given that the role of P2RX7 on cancer cell death and proliferation has been widely studied [8], this review will mainly focus on the role of P2RX7 in modulating the antitumor immune response and its potential to constitute a novel antitumoral strategy.

## 2. P2RX7: A Unique P2X Receptor with Several Activities Impacting Tumor Growth

P2RX7 belongs to the family of P2X receptors that are assembled and active when in their trimeric form. Each monomer is composed of two transmembrane domains that are connected by a large extracellular loop, and an N- and C- termini domain located intracellularly. However, unlike other members, P2RX7 has a long intracellular C-terminal domain that structurally distinguishes it from the others and confers its unique biological activities.

Even though all seven members of the P2X receptors recognize eATP, they are activated with various affinities that range from 0.5 µM for P2RX3 to over 100 µM for P2RX7 [9]. Thus, activation of P2RX7 requires high levels of eATP, levels that are found in the TME [10] which controls the three main activities of the receptor: cationic exchange, macropore opening and NLRP3 inflammasome activation.

### 2.1. Cationic Exchange

P2RX7, as well as the other P2X receptors, are non-selective cationic channels that lead to membrane depolarization when bound to eATP. Indeed, activation of P2RX7 leads to calcium and sodium influx as well as potassium efflux, which ultimately activate various signaling pathways that range from cell proliferation to immune-related events [11].

It has been reported that low levels of eATP can induce cell proliferation in a P2RX7-dependent manner through calcium influx [8]. Indeed, intracellular calcium is a second messenger that controls various cellular functions. It has been shown that the ATP/P2RX7 axis leads to the activation of MAP kinases (ERK1/2, p38, JNK), B kinases (PKB, Akt) and PKC involved in cell proliferation in several types of cancer, in mouse and human models [8,11]. Moreover, increase of intracellular calcium levels following P2RX7 activation has been shown to promote cancer cell survival via mitochondrial ion homeostasis [12], calcineurin activation and NFAT translocation into the nucleus [13].

P2RX7-dependent calcium influx has also been shown to be implicated in the immune response. P2RX7 stimulation in mouse microglial cells induced NFAT and NF-κB translocation into the nucleus in a calcineurin-dependent manner, and both are required for the expression of inflammatory cytokines [14] and chemokines such as CCL3 [15]. On the other hand, the increase of intracellular calcium and extracellular potassium concentrations have been shown to limit B cell activation by decreasing NFAT translocation into the nucleus [16]. P2RX7 has also been shown to have a role in immune cell proliferation [17]. Indeed, the activation of the ATP/P2RX7 axis in human T cells induces NFAT translocation and IL-2 secretion, necessary for T cell survival and proliferation [18]. Accordingly, it has been shown that the expression of P2RX7 on mouse CD8+ T cells enhances mitochondrial functions through calcium influx and AMPK activation, required to support the generation and survival of memory T cells [19] and efficiently eradicate tumor cells [20]. In line with these findings, the same authors have shown, in viral infections mouse models, that P2RX7 induces the expression of the TGBRII receptor through calcineurin activation, thereby sustaining the generation of tissue-resident memory T cells [21] that require TGFβ sensing for their survival. However, a contradictory study had previously shown, in a mouse model of melanoma, that the expression of P2RX7 on tumor-infiltrating lymphocytes has a detrimental role by limiting T cell proliferation and inducing T cell senescence through mitochondrial reactive oxygen species (ROS) and p38 MAPK activation [22]. The discrepancy between the studies could be due to a different activation protocol of T cells, as explained by the authors [20].

### 2.2. Macropore Opening

One particular characteristic of P2RX7 is the capacity to induce cell permeabilization and death, which paved the way to the discovery of the receptor. The receptor was initially named P2Z since high levels of ATP were shown to induce cell death in human macrophages [23]. Its cloning in 1996 highlighted that its structure is close to the P2X receptors [24], which led to it becoming the seventh member of the P2X family.

The P2RX7-dependent cell permeabilization is linked to its ability to cause the opening of macropores of 8.5 Å on the plasma membrane, leading to cell death [25]. The pores allow the passing of macromolecules up to 900 Da in a non-selective and bidirectional manner, disturbing the intracellular homeostasis [25]. The nature of this pore is still under debate. Either (1) P2RX7 recruits pore-forming proteins following its activation, such as pannexine-1 or connexine-43 [26,27,28], or (2) the pores are shaped due to the dilatation of the canal formed by the three monomers of P2RX7 [29,30,31,32]. Nevertheless, it is certain that the pore-forming activity of P2RX7 requires the presence of 177 aminos acids in the C-terminal domain [24], which explains the selective activity of P2RX7, as compared to the other P2X receptors.

Given its ability to induce cell death, one can speculate that the expression of P2RX7 on tumor cells is beneficial. Indeed, studies using tumor cell lines and preclinical mouse models in melanoma [33,34], non-melanoma skin cancers [35], intestinal carcinomas [36] and breast cancer [37] have shown that activation of P2RX7 reduced cell proliferation by inducing cell death. On the other hand, many reports show an increase of P2RX7 expression in various tumor cell lines and in patients [38,39,40,41,42] and this increase is linked to tumor growth, as discussed in the previous section. When the human *P2RX7* gene is subjected to alternative splicing, a C-terminal-deleted variant named P2RX7-B [43] is generated, preventing the receptor from forming macropores. This accords with reports that P2RX7-B is overexpressed in lung adenocarcinomas [44], acute myeloid leukemia [45], osteosarcoma [46], neuroblastoma [47] and glioblastoma [48]. In addition, a non-functional form of P2RX7 that lacks macropore activity has been reported in several types of cancers including lung and glioblastomas [49,50]. Whether the expression of the non-functional form is linked to P2RX7-B is currently unknown. However, the existence of isoforms unable to form macropores can explain how a cell death-inducing receptor can favor tumor growth.

Not only is the macropore activity associated with tumor cell death, but also with immune functions. P2RX7 is implicated in the regulation of T cell homeostasis through induction of cell death, which is linked to the levels of expression of the receptor. Indeed, the expression levels of P2RX7 are regulated according to the T cell subset, its activation status as well as its localization as extensively reviewed in [51,52]. For instance, it has been shown that the expression of P2RX7 is downregulated following the activation of T cells, protecting them from cell death [53]. This is the case for follicular T cells in Peyer’s Patches [54] and tissue-resident memory T cells [55,56]. The purpose of P2RX7’s downregulation is not only to prevent T cell death by protecting antigen-reactive T cells, but also to favor the ionic activity of the receptor and to confer a better fitness of CD8+ T cells [19], which consequently enhances the control of tumor growth [20].

On the other hand, T regulatory cells (Tregs) and natural killer T cells (NKT) express high levels of P2RX7 and are more susceptible to P2RX7-induced cell death [51,57]. We have shown that *p2rx7-/-* mice with colitis-associated cancer develop bigger and more aggressive lesions than WT mice, which was associated with an accumulation of Tregs within the lesions [58]. Moreover, early depletion of Tregs in vivo by activation of P2RX7-expressing T cells using the NAD+/ART2 axis (known to activate P2RX7 in mouse T cells [53]) in tumor mouse models increased the anti-tumor effector functions of CD8+ T cells [59].

### 2.3. NLRP3 Inflammasome Activation

Another feature of P2RX7 that differentiates it from other P2X receptors is its ability to activate the NLRP3 inflammasome. The NLRP3 (NOD-like receptor family, pyrin domain containing 3) protein belongs to the NOD-Like receptors (NLR) family which are pattern recognition receptors (PRR), meaning that NLRP3 is implicated in danger signal recognition [60], highlighting its importance in the establishment of the immune response, notably in the context of cancer. It is, therefore, mainly studied in the innate immune cells from the myeloid lineage such as macrophages, monocytes and dendritic cells. This is supported by the high expression levels of P2RX7 on these cells compared to other immune cells [11,61].

Even though NLRP3 is not a receptor per se, it is a cytosolic sensor of stress that interacts with other proteins to form a multimeric complex of proteins called the NLRP3 inflammasome. The complex involves the NLRP3 protein (containing a pyrin domain), the apoptosis-associated speck-like protein containing a CARD (ASC, with a CARD and pyrin domain) and the effector protein pro-caspase-1 (with a CARD domain). The proteins associate by way of ASC, which acts as an anchor through the respective binding of the CARD and pyrin domains of NLRP3 and pro-caspase-1. The NEK7 protein (NIMA-related kinase 7) is necessary for the oligomerization of the NLRP3 subunits to form the NLRP3 inflammasome [62]. The oligomerized NLRP3 inflammasome causes the self-cleavage of the pro-caspase-1 into active caspase-1, which, in turn, cleaves the inactive precursors of the IL-1β and IL-18 inflammatory cytokines intracellularly to shape the immune response.

The ATP/P2RX7 axis is not the only activator of the NLRP3 inflammasome but is described as the most potent [63]. Indeed, the NLRP3 inflammasome can also be activated by a variety of stress signals such as pore-forming toxins, ionophores or uric acid crystals [64]. It has been reported that the main event leading to the assembly of the NLRP3 inflammasome is potassium efflux, which is common for all its activators, including ATP [64,65]. Potassium efflux is a common feature of all the P2X receptors as well as other potassium channels, yet they are unable to independently activate the NLRP3 inflammasome. Indeed, the C-terminal domain of P2RX7 is necessary for the activation of NLRP3 by ATP, since its absence prevents NLRP3 activation and IL-1β release [32]. Moreover, unlike the other P2X receptors, P2RX7 does not desensitize [9], leading to continuous potassium efflux that may facilitate NLRP3 activation. On the other hand, NLRP3 and P2RX7 have been shown to interact at the plasma membrane either directly [66] or indirectly via the Paxillin protein [67] to induce the assembly of NLRP3. It has also been shown that P2RX7-mediated NLRP3 activation requires the potassic channel TWIK2, where both receptors act in synergy to decrease intracellular potassium levels and activate NLRP3 [68]. Besides potassium efflux, P2RX7-mediated calcium influx has also been shown to assemble NLRP3, through ROS production and mitochondrial depolarization [69]. Therefore, the potency of P2RX7 to activate NLRP3 could be explained by the fact that P2RX7 can induce calcium influx, potassium efflux and interact with other potassic proteins as well as with NLRP3 itself, leading to a robust activation of the NLRP3 inflammasome.

The role of NLRP3 in cancer progression is not clear. Indeed, there are several conflicting studies in various types of cancer showing that NLRP3 can either favor tumor growth or act as a tumor suppressor [70,71]. Since the readout activity of NLRP3 is the release of mature inflammatory cytokines (Figure 1), we will focus in the next section on the role of IL-1β and IL-18 in cancer progression.

## 3. IL-1β and IL-18: The Main Players in the Immunomodulation by P2RX7

Even though the activation P2RX7 induces a plethora of immune-related events such as T cell activation via the shedding of CD62L [51], the control of phagocytosis [72] and the release of cytokines [73,74], the main and most studied function of P2RX7 is its ability to release IL-1β and IL-18.

### 3.1. Biology and Signaling of IL-1β and IL-18

As mentioned before, the IL-1β and IL-18 cytokines are released from the cell after the cleavage of their precursors (pro-IL-1β and pro-IL-18, respectively) by caspase-1. However, a priming step prior to the activation of P2RX7 is required, involving the activation of TLR4/NF-κB pathway by Pathogen Associated Molecular Patterns (PAMPs). Not only does this priming step allow post translational modifications on NLRP3 for an optimal assembly [75,76], but it is also crucial in inducing the expression of pro-IL-1β and upregulating the expression of the other components of the inflammasome. Unlike pro-IL-1β, the expression of pro-IL-18 is constitutive, as seen in immune cells [77] and in epithelial cells and pulmonary fibroblasts [78,79]. It was reported in the past that the activation of the NLRP3 inflammasome cannot occur without the priming step; however, it has recently been shown that it can be dispensable in the release of IL-18 from human monocytes [80].

Apart from the fact that the differential expression of the cytokines contributes to the regulation of their activity, both cytokines are regulated on another level that involves their receptors and antagonists, as discussed hereafter.

#### 3.1.1. IL-1β

IL-1β is recognized by the IL-1R1 receptor. The binding of IL-1β to its receptor triggers the recruitment of the IL-1RAcP adaptor protein to form an IL-1β/IL-1R1/IL-R1AcP complex required for the activation of the downstream MYD88/NF-κB signaling. Another cell surface receptor called IL-1R2 can also bind IL-1β with higher affinity than IL-1R1, but is a decoy receptor. IL-1R2 prevents the recruitment of the MYD88 protein and the activation the NF-κB pathway, since it lacks the intracellular TIR domain required for MYD88 recruitment. The IL-1β signaling can also be inhibited by another antagonist called IL-1Ra that binds the IL-1R1 receptor and blocks the association of IL-1RAcP to the complex [81,82,83].

#### 3.1.2. IL-18

IL-18 signaling is similar to that of IL-1β. IL-18 binds to the IL-18Rα (IL-18R1) subunit recruiting the high affinity IL-18Rβ (IL-18RAP) subunit for MYD88/NF-κB activation. However, IL-18Rβ is only expressed by T cells, NK cells and dendritic cells [84]. Indeed, IL-18 was initially named IGIF (IFN-γ inducing factor) [85], as this cytokine induces IFN-γ production by T cells and NK cells after its binding to the IL-18Rα/IL-18Rβ complex. IL-18 binds, however, with higher affinity to its natural antagonist called IL-18 Binding Protein (IL-18BP). IL-18BP is constitutively secreted in high amounts to prevent the binding of IL-18 to its receptor and acts as a negative feedback loop to IL-18 activity, since its expression is increased when high levels of IFN-γ are produced [84].

As IL-1β was discovered long before IL-18, its role in inflammation and cancer progression has been vastly studied and well-reviewed [70,86,87,88]. IL-1β has been shown to have both pro- and anti-tumor effects. For instance, it is known that IL-1β can favor tumor growth by inducing an immunosuppressive environment through the recruitment of myeloid-derived suppressors cells (MDSCs) and tumor-associated macrophages (TAMs) [89,90], as well as by favoring angiogenesis [91,92]. On another hand, IL-1β has also been shown to shape the T cell response and favor antitumor immunity through the ATP/P2RX7/NLRP3 axis in dendritic cells [93,94]. As IL-18 has recently attracted attention in tumor biology, and the implication of IL-1β was extensively discussed elsewhere, the rest of this review will focus on the role of IL-18 in cancer progression.

### 3.2. IL-18: An Antitumoral Cytokine

The role of IL-18 has been studied in several types of cancer. Preclinical studies have shown that IL-18 is required to inhibit tumor growth using either *il18-/-* or *il18r1-/-* mice, or even recombinant IL-18 administration.

In AOM/DSS-induced colon cancer, it has been shown that IL-18 levels were decreased in tumor-bearing mice [95], that the NLRP3/ASC/CASP1/IL-18 axis was important to dampen tumor growth [95,96] and this was dependent on IFN-γ production [97]. Indeed, there is an absence of the components of the NLRP3 inflammasome, IL-18 exacerbated cancer progression [97] and inflammatory markers [98]. In line with this, systemic administration of IL-18 in the same model and in the MC38 subcutaneous model reduced disease aggressiveness as well as the cytotoxicity of NK cells, CD4+ and CD8+ T cells [96,99,100,101], thus, demonstrating the ability of IL-18 to enhance antitumor immunity by boosting IFN-γ production. Moreover, IL-18 production and administration was shown to limit the migration of colon MC38 tumor cells into the liver by increasing the activation of NK cells and their cytotoxicity through FASL [102]. Similar results have been shown in pancreatic, metastatic and non-metastatic melanoma mouse models, where in vivo administration of IL-18 slowed tumor growth by increasing the activation and cytotoxicity of CD4+ and CD8+ T cells and NK cells [101,103,104,105], as well as their interaction with tumor cells and the robust generation of memory T cells [106]. It has also been recently shown that IL-18 administration after bone marrow transplantation induced the cytotoxicity of T cells in a mouse model of myeloma and enhanced their lethality in a leukemia model [107].

The role of the ATP/P2RX7/NLRP3 axis in IL-18 release and antitumor immunity was investigated. We have shown that activation of P2RX7 using a positive modulator (HEI3090) inhibits tumor growth in lung and melanoma mouse models. This effect relies on IL-18 production by dendritic cells to boost antitumor immunity, e.g., IFN-γ production by CD4+ T cells and NK cells [85,103]. Moreover, using blocking antibodies against the ectonucleotidases responsible for ATP degradation, namely CD73 (or NT5E) and CD39 (or ENTPD1), antitumor immunity was boosted through the P2RX7/NLRP3 pathway, and tumor growth was inhibited in mouse models of melanoma, fibrosarcoma, colon and prostate tumors [108,109]. It is interesting to note that these studies have also shown that IL-18 can synergize with αPD-1 immunotherapy or chemotherapy for stronger antitumor immunity and tumor growth inhibition. Indeed, activation of the P2RX7/IL-18 axis with HEI3090 sensitized lung and melanoma tumors to immunotherapy, cured 80% of mice and protected them from a tumor rechallenge, highlighting the potency of IL-18 in generating long lasting memory T cells in tumor models [110]. The ability of IL-18 to generate memory T cells is mainly studied in infection models, where IL-18R is shown to be upregulated [111,112]. This is supported by the fact that the release of IL-18 in DC and the IL-18R/MyD88/IFN-γ axis in T cells during T cell activation favors the expansion of memory T cells [113,114,115]. Interestingly, IL-18 can also induce IFN-γ production even in absence of TCR stimulation [111,112]. These studies underly the power of IL-18 in enhancing antitumor immunity through T cell cytotoxicity and T memory cell formation.

IL-18 increases the activation and cytotoxicity of T cells and NK cells, as evidenced in tumor models. In vitro activation of effector NK and T cells before their transfer in vivo has shown powerful antitumor effects and constitute a promising strategy. Transfer of in vitro pre-activated NK cells with IL-18, IL-12 (another inducer of IFN-γ) and IL-15 (activator of NK cells) in a melanoma mouse model under radiation therapy enhanced antitumor immunity and decreased lung metastases, by increasing the accumulation and cytotoxicity of in vivo NK cells that required the presence of CD4+ T cells [116]. The same efficacy in NK cell cytotoxicity was observed in human NK cells derived from patients with metastatic melanoma [117] and healthy subjects, and this enhanced the killing of human colon, melanoma, glioblastoma, prostate, breast and ovarian tumor cell line spheroids [118]. Due to the powerful antitumor potential of IL-18, it has been used in preclinical studies as a tool to increase the proliferation, survival and cytotoxicity of CAR T cells (Chimeric Antigen Receptor T cells) in melanoma [119,120,121,122,123,124] and small cell lung cancer [125].

The antitumor potential of IL-18 has also been evaluated in cancer patients. Levels of IL-18 and IFN-γ were shown to be lower in plasmas of non-small cell lung cancer patients [126] and in tumor tissue of colon cancer, melanoma, thyroid and esophageal cancer patients compared to those of healthy subjects, and low levels of IL-18 correlated with the presence of metastases, a poor outcome [127,128,129,130,131] and increased risk of developing acute myeloid leukemia [132]. Moreover, IL-18 levels were shown to be correlated with higher CD8+ T cells, NK cells infiltration and activity in colorectal cancer [133], melanoma [129] and thyroid carcinoma [130]. Accordingly, functional intratumoral CD8+ T cells were shown to express high levels of IL-18R in non-small cell lung cancer patients and were, furthermore, cytotoxic in the presence of IL-18 [126]. IL-18R was clearly shown to be correlated to immune infiltrate in lung squamous carcinoma and to improved overall survival [134]. However, IL-18BP levels have been shown to be high in sera of patients with ovarian [135], non-small cell lung cancer [101] and prostate cancer [136], as well as in tumor tissue of melanoma, breast, head and neck, gastric, ovarian and prostate cancers [101,135,136]. These findings are in alignment with the observation that high levels of IL-18 in sera and tissue of ovarian cancer patients were shown to be inactive given the lack of IFN-γ production [137]. This is consistent with the fact that IL-18 signaling is linked to better antitumor response and survival. Indeed, it was shown that after the first round of αPD-L1 immunotherapy, IL-18 levels increase in sera of patients with multiple myeloma, non-small cell lung cancer, renal cell carcinoma and melanoma, which is linked to a reactivation and cytotoxicity of antitumor effector cells, namely IFN-γ production [138,139].

Given the potency of IL-18 to trigger the activation and cytotoxicity of immune effector cells and its ability to enhance the efficacy of immunotherapy, and given the lack of biomarkers to predict patients’ response to immunotherapy, recent studies have investigated the potential of IL-18 in predicting patient outcome. High levels of IL-18 were shown to be found in plasmas and tissue of immunotherapy-responding non-small cell lung cancer patients, and were linked to an antitumor immune gene signature [140] and a reduced tumor burden [139,141].

### 3.3. IL-18: A Pro Tumoral Cytokine

Even though IL-18 has extensively been shown to boost antitumor immunity and to be associated with a better outcome in both murine and human settings, several studies have pointed out a pro-tumoral role of IL-18.

Indeed, IL-18 is associated with increased migration of cancer cells and metastatic activity. In vitro studies have shown that the migration ability and proliferation of human leukemic cells, lymphoma [142,143], human gastric cells and murine melanoma was dependent on their production of IL-18 in a NLRP3/caspase-1-dependent manner through VEGF production [144,145,146], and this was further increased when cells were treated with recombinant IL-18 [147]. Furthermore, in vivo neutralization of IL-18 with IL-18BP reduced melanoma metastases into the liver in a mouse model [148] and *il18-/-* mice exhibited less tumor growth in a multiple myeloma mouse models [149]. In multiple myeloma, the pro-tumoral effect of IL-18 was shown to be due of its release in a NLRP3-dependent from TAMs [150] and favored the immunosuppressive activity of MDSCs [149]. Not only levels of IL-18 were increased in tumor tissue of mouse models of pancreatic cancers [151,152], it was also shown to have a pro-tumoral activity. Indeed, IL-18 release is dependent on NLRP3 activation, and its signaling through the IL-18R receptor induces the exhaustion of CD8+ T cells [153] and limited migration into the tumor [154]. This is supported by another study in melanoma and colon tumor mouse models showing that IL-18 induces the expression of PD-1 on NK cells [155]. An earlier study had shown that NF-κB-dependent expression of IL-18 by human and murine pancreatic tumor cells increased their proliferation and invasion in vitro and in vivo, and they were blocked in the presence of IL-18BP. However, the authors also showed that IL-18 enhanced the cytotoxicity of NK cells and T cells, making IL-18 a cytokine with both pro- and antitumoral proprieties. The combination of NF-κB inhibition and IL-18 administration to mice significantly reduced tumor growth, taking advantage of both properties of the cytokine [152]. IL-18 would have a different effect depending on the cell type, which demonstrated by the facts that human pancreatic tumor tissue expresses high levels of IL-18 [152,153,156,157] and low levels of IL-18BP [156] and these are linked to shorter survival and increased metastasis, whereas high levels of plasmatic IL-18 are associated with a better outcome [152]. Similarly, NLRP3-associated IL-18 is found in high levels in plasma of lymphoma [142] and multiple myeloma patients where IL-18 is a predictor of poor survival in multiple myeloma patients [149,158], esophageal carcinoma [159], renal cell carcinoma [160] and gastric carcinoma [161]. High expression of IL-18 is also associated with increased metastases in hepatocellular and breast carcinomas [162,163]. As for the role of IL-18 in predicting patients’ response to immunotherapy, it was shown that patients with non-small cell lung cancer who respond to treatment had lower levels of IL-18 compared to non-responders [164], which contrasts with other studies in the same tumor type [140,141].

## 4. The P2RX7/IL-18 Pathway: A Target in Cancer?

P2RX7 can be considered to be a fine tuner of the tumor microenvironment, given that it is able to induce cell proliferation and death as well as to modulate the immune response through various pathways, notably through the release of IL-18 following the activation of the NLRP3 inflammasome.

Even though P2RX7 has been shown to promote tumor growth when expressed by tumor cells [8], its expression and activity on immune cells has mostly been associated with antitumor effects by enhancing antitumor immunity. As described by us and others, this effect relies on the NLRP3-dependent release of IL-18 by dendritic cells, which enhances the cytotoxicity of antitumor effector cells. This is in agreement with other work showing a strong antitumor activity of IL-18 as well as its potential as a biomarker for tumor progression and patient outcomes. However, several other studies pointed out a pro-tumoral role of IL-18.

The dual roles of IL-18 could have several explanations. IL-18 could have a pro-tumoral activity in certain types of cancers, such as in pancreatic cancer and multiple myeloma, as discussed in this review, and the source and frequency of IL-18 release may contribute to its effect. Indeed, it was shown that daily systemic administration of rIL-18 had antitumor effects, whereas once weekly or bi-weekly administration favored lung metastases of melanoma cells and induced a strong immunosuppressive environment [155], hinting that the levels of IL-18 are critical for its effect. The pro-tumoral role of IL-18 seems also to be linked to its release from tumor cells, whereas its antitumoral role seems mainly be linked to its release from dendritic cells, therefore, shaping the immune response. This supports the predictive value of plasmatic IL-18, since it can be released from tumor and immune cells. In line with this, it has been shown that the processing of IL-18 from ovarian tumor cells can be defective [165,166], which could explain its pro-tumoral activity when released from tumor cells. Only a few studies have addressed the activity of IL-18, i.e., the free, unbound IL-18 from IL-18BP. Indeed, IL-18BP is present physiologically at high levels and has been shown to be further increased in several cancers, inhibiting even more IL-18 signaling. Therefore, determining the levels of free, unbound IL-18 is key to understanding its role. It should be cautioned that the exact role of IL-18 can be misinterpreted when looking at mRNA levels. Since it is constitutively expressed and requires a proteolytic cleavage to be released and active, looking at the proteic levels of IL-18 would be more meaningful. Additionally, the pro-tumoral role of IL-18 seems to be linked to T cell and NK cell exhaustion. This could be beneficial when combined with immune checkpoint inhibitors. This was in fact the case when using HEI3090 or anti-CD39/CD73 or even DR-IL-18 (a decoy resistant IL-18) [101,108,109,110]. The exact role of IL-18 in cancer progression warrants further investigation to determine (i) the cancer type, (ii) the cell source and (iii) its activity (free unbound IL-18).

In any case, the studies have been consistent in showing that IL-18 release in an immune-P2RX7-dependent manner is highly antitumoral, and suggesting that boosting the P2RX7/IL-18 pathway is promising for several types of cancer. However, one cannot exclude the studies that show P2RX7 itself has also pro-tumoral effects. Even though we and others have shown in vivo that boosting the activation of P2RX7 occurs mostly in immune cells [108,110], caution should also be taken with the functionality of the receptor expressed by tumor cells.

## 5. Conclusions

Preclinical studies point towards strong antitumor efficacy of the P2RX7/IL-18 axis in several types of cancers. Targeting the P2RX7/IL-18 axis is therefore promising in humans and encourages further studies to confirm these results in human settings.

## Figures and Tables

**Figure 1 ijms-24-09235-f001:**
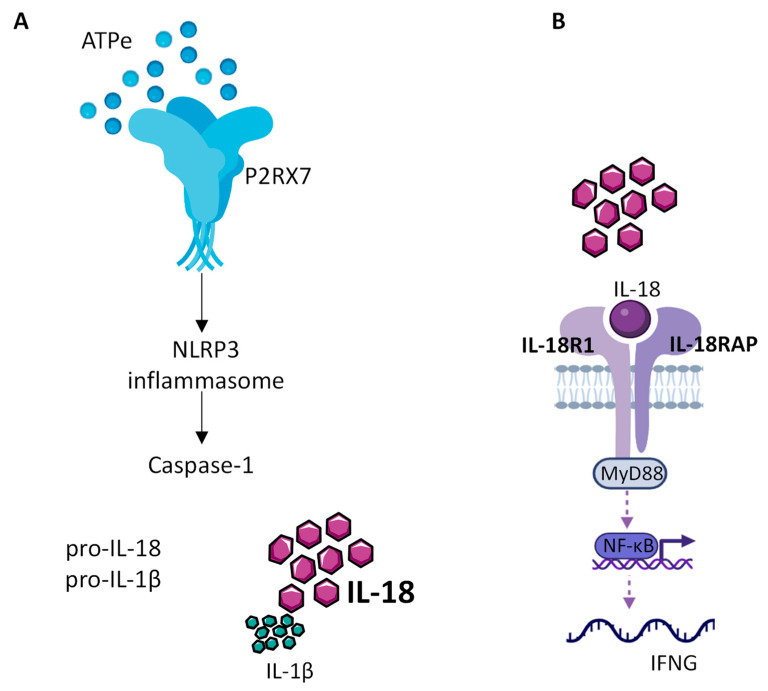
IL-18 production by the NLRP3 inflammasome. (**A**) Extracellular ATP (eATP), present at high concentration within the TME, activates P2RX7, which in turn allows the assembly and activation of the NLRP3 inflammasome leading to caspase1 activation, cleavage of the constitutive pro IL-18 cytokine and release of mature IL-18. (**B**) Released mature IL-18 binds to IL-18 receptor, composed of IL-18R1 and IL-18RAP, to activate NF-kB, which in turn controls the production of INF-γ.

## Data Availability

Not applicable.

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
