# Peer review of "The Role of IL-18 in P2RX7-Mediated Antitumor Immunity"

_ijms, 2023, doi:10.3390/ijms24119235_

Round 1
Reviewer 1 Report
This is a thorough and timely addition to the literature on NLRP3/IL-18 effects in tumor biology. Authors carefully delineate pro- and anti-tumor mechanisms. It is understandable and sound that - based on their own studies - they lean into the antitumoral effects of P2RX7 and NLRP3 activation. I have only minor comments:
- comprehensive discussion of the influence of IL-18 on T cell plasticity should discusses the role of NLRP3/IL-18 for T cell memory formation. It would then become clear that T cell exhaustion is just one aspect of the broader molecular mechanisms of T cell plasticity influenced by IL-18.
- the dichotomal role of NLRP3 activation in cancer progression is similar to its role in inflammatory models, e.g. colitis. This might be interesting as it further emphasises the role of the TME and cooperation of various immune cell populations.
- in many studies the cellular source of IL-18 remeinas unclear. The authors could use this review to analyse the literature systematically on what is known about the producer cells of IL-18 (tumor, dendritic cells, macrphages), also what is known about the molecular events resulting in IL-18 secretion in the various tumor biology studies.
- possibly a title including "IL-18" or "NLRP3" would result in a wider readership that catches interest when screening the literature.
The level of English is very good. In line 24, "improved disease aggressiveness" is a bit of an oxymoron.
Author Response
Comments and Suggestions for Authors
This is a thorough and timely addition to the literature on NLRP3/IL-18 effects in tumor biology. Authors carefully delineate pro- and anti-tumor mechanisms. It is understandable and sound that - based on their own studies - they lean into the antitumoral effects of P2RX7 and NLRP3 activation. I have only minor comments:
Thank you for your comment.
- comprehensive discussion of the influence of IL-18 on T cell plasticity should discusses the role of NLRP3/IL-18 for T cell memory formation. It would then become clear that T cell exhaustion is just one aspect of the broader molecular mechanisms of T cell plasticity influenced by IL-18.
We thank the reviewer for this suggestion; the role of IL-18 in T cell memory formation is in fact very interesting. We have added the corresponding discussion in the review:
The ability of IL-18 to generate memory T cells is mainly studied in infections models, where IL-18R is shown to be upregulated [112,113]. This is coherent with the fact that the release of IL-18 in DC and the IL-18R/MyD88/IFN-γ axis in T cells during T cell activation favors the expansion of memory T cells [114–116]. Interestingly, IL-18 can also induce IFN-γ production even in absence of TCR stimulation [112,113]. These studies underly the power of IL-18 in enhancing antitumor immunity through T cell cytotoxicity and T memory cell formation.
- the dichotomal role of NLRP3 activation in cancer progression is similar to its role in inflammatory models, e.g. colitis. This might be interesting as it further emphasises the role of the TME and cooperation of various immune cell populations.
We completely agree. The dichotomic role of NLRP3 in cancer and inflammation is not very surprising since sustained inflammation favors cancer progression as extensively studied in colitis. As in cancer, NLRP3 has been shown to have both pro-inflammatory and anti-inflammatory roles. One can speculate that it could be due to the different microbiota of mice used in the various studies that may trigger a different inflammatory response. The cell source of NLRP3 activation as well as the timing of its activation during the experimental model may also influence the impact of NLRP3 activation. This highlights indeed the crucial role of the microenvironment in the outcome of NLRP3 activation. However, we have the feeling that this particular point is out of the scope of this review and that this very interesting subject has to be discussed in a dedicated paper.
- in many studies the cellular source of IL-18 remeinas unclear. The authors could use this review to analyse the literature systematically on what is known about the producer cells of IL-18 (tumor, dendritic cells, macrphages), also what is known about the molecular events resulting in IL-18 secretion in the various tumor biology studies.
We agree that a systematic description of IL-18-producer cells is an interesting subject as well as the molecular events resulting in IL-18 secretion. However, we thought that is would be more appropriate in this review to discuss what is known regarding the role of IL-18 signaling pathway (i.e., the release of IL-18 and the expression of its receptors) in tumor biology and we opted to described studies highlighting the role of IL-18 as an antitumor cytokine (in 3.2) or a protumor cytokine (in 3.3).
- possibly a title including "IL-18" or "NLRP3" would result in a wider readership that catches interest when screening the literature.
The we do agree that IL-18 and NLRP3 are two keywords of this review alongside P2RX7, it is why we had included IL-18 in the title. Given this comment we speculated that the emphasis on IL-18 is not enough therefore we propose the following title:
The role of IL-18 in P2RX7-mediated antitumor immunity
Comments on the Quality of English Language
The level of English is very good. In line 24, "improved disease aggressiveness" is a bit of an oxymoron.
Thank you for your comment. We have changed the word improved with “reduced” as follow:
In line with this, systemic administration of IL-18 in the same model and in the MC38 subcutaneous model reduced disease aggressiveness as well as the cytotoxicity of NK cells, CD4+ and CD8+ T cells [97,100–102], showing the ability of IL-18 to enhance antitumor immunity by boosting IFN-γ production.
Reviewer 2 Report
I thought this review was very well organized regarding P2X and NLRP3. I would appreciate your consideration of the following points
1) If you do not consider IL1 important for the content of this review, why not remove 3.1.1?
2) It has been admitted that low levels of eATP can induce cell proliferation in a P2RX7- 56
dependent manner through the calcium influx.
Please add a citation to this statement.
3) Is P2RX7 activation involved in IL18 secretion without NLRP3 activation?
4) How much has been said about the association between other Inflammasome and tumors, I understood that if IL18 is important, it may inhibit tumor growth as well.
5) I suggest you create a figure that shows this flow of P2RX7⇨NLRP3 inflammasome→IL18.
Author Response
Comments and Suggestions for Authors
I thought this review was very well organized regarding P2X and NLRP3. I would appreciate your consideration of the following points
We thank the reviewer for his comment.
- If you do not consider IL1 important for the content of this review, why not remove 3.1.1?
Of course, we do not consider that IL-1b is not important, as evidence by the numerous reviews discussing its role in tumor biology (PMID 32635472, PMID 30042333). In this review we decided to focus the less described role of IL-18, in particular on the role of the eATP/P2RX7/NLRP3/IL-18 pathway in tumor biology. However, it would have been difficult not to mention IL1b downstream of this pathway knowing that this is the most described cytokine downstream of NLRP3 in the literature. To clarify this point, we now wrote:
As IL-18 has recently attracted attention in tumor biology and the implication of IL-1β was extensively discussed elsewhere, the rest of this review will focus on the role of IL-18 in cancer progression.
2) It has been admitted that low levels of eATP can induce cell proliferation in a P2RX7- 56 dependent manner through the calcium influx.
Please add a citation to this statement.
The citation has been added, sorry for the omission.
3) Is P2RX7 activation involved in IL18 secretion without NLRP3 activation?
The main event leading to the release of IL-18 is dependent on the activation of the NLRP3 inflammasome. Very few studies have shown that other proteases than caspase-1 can cleave pro-IL-18. The granzyme B, produced by NK cells and CD8+ T cells, and the chymase, produced by mast cells, have been shown to cleave pro-IL-18 at various sites other than the one for caspase-1. This could explain the fact that the mature form of IL-18 generated by these proteases is much less potent than the caspase-1-processed IL-18 (PMID: 26084020)
P2RX7 has been shown to be implicated in mast cell degranulation that leads to the release of proteases that include the chymase (PMID: 26910735). On the other hand, granzyme B levels were shown to depend on the expression of P2RX7 on mouse T cells (PMID: 32699136, PMID: 36248812). However, to our knowledge, no study has addressed the implication of P2RX7 in the granzymeB/chymase-dependent release of IL-18.
Given the very few studies on this subject, we opted not to include this topic in this review even though it is extremely interesting.
4) How much has been said about the association between other Inflammasome and tumors, I understood that if IL18 is important, it may inhibit tumor growth as well.
It is true that activation of all inflammasomes lead to the release of IL-1β and IL-18. Our review focuses on the role of IL-18 following P2RX7 activation. Since NLRP3 is the only described inflammasome to date to be activated by P2RX7, we only addressed NLRP3 in the review.
One can in fact speculate that IL-18 release following the activation of other inflammasomes has also antitumor effects; this has been indeed shown for NLRP1, AIM2 and NLRP6 in colorectal mouse models PMID: 25725098 PMID: 27524110 PMID: 21543645. However, studies have also shown that all the inflammasomes, including NLRP3, have dual roles in cancer progression, as recently reviewed in PMID: 36932407 and PMID: 37020871.
5) I suggest you create a figure that shows this flow of P2RX7⇨NLRP3 inflammasome→IL18.
As suggested by the reviewer, we have included a figure to illustrate the production of IL-18 downstream of the eATP/P2RX7 pathway.